# Looking Back, Looking Forward: A Study Protocol for a Mixed-Methods Multiple-Case Study to Examine Improvement Sustainability of Large-Scale Initiatives in Tertiary Hospitals

**DOI:** 10.3390/healthcare11152175

**Published:** 2023-07-31

**Authors:** Sarah E. J. Moon, Anne Hogden, Kathy Eljiz, Nazlee Siddiqui

**Affiliations:** 1Australian Institute of Health Service Management, College of Business and Economics, University of Tasmania, Sydney 2040, Australia; 2Statewide Quality and Patient Safety Service, Department of Health Tasmania, Launceston 7250, Australia; 3School of Population Health, Faculty of Medicine and Health, University of New South Wales, Sydney 2052, Australia

**Keywords:** organizational change, development, change sustainability, quality, improvement, hospital, patient safety, organizational learning, change management, knowledge translation

## Abstract

*Background* Hospitals invest extensive resources in large-scale initiatives to improve patient safety and quality at an organizational level. However, initial success, if any, does not guarantee longer-term improvement. Empirical and theoretical knowledge that informs hospitals on how to attain sustained improvement from large-scale change is lacking. *Aim* The proposed study aims to examine improvement sustainability of two large-scale initiatives in an Australian tertiary hospital and translate the lessons into strategies for achieving sustained improvement from large-scale change in hospital settings. *Design and Methods* The study employs a single-site, multiple-case study design to evaluate the initiatives separately and comparatively using mixed methods. Semi-structured staff interviews will be conducted in stratified cohorts across the organizational hierarchy to capture different perspectives from various staff roles involved in the initiatives. The output and impact of the initiatives will be examined through organizational documents and relevant routinely collected organizational indicators. The obtained data will be analyzed thematically and statistically before being integrated for a synergic interpretation. *Implications* Capturing a comprehensive organizational view of large-scale change, the findings will have the potential to guide the practice and contribute to the theoretical understandings for achieving meaningful and longer-term organizational improvement in patient safety and quality.

## 1. Introduction

Making and maintaining improvements is vital for hospitals to deliver safer and higher-quality care. Hospitals face escalating public scrutiny and regulatory requirements for safer and higher-quality care [1]. In response, hospitals dedicate resources to implementing change for improvement, such as quality improvement initiatives [2,3,4]. Large-scale quality initiatives, such as those implemented at an organizational level in hospitals, typically entail mobilizing larger groups of stakeholders whose priorities may diverge or compete. A successful introduction of change necessitates the concerted efforts of those involved at various levels of the organization as well as more time and resource investments. Additionally, achieving the stability of these achievements over the long term cannot be guaranteed by any initial success. Consequently, it poses a major challenge to attain sustained improvement through time and resource intensive endeavors with multiple stakeholder engagements. Based on previously established definitions [5,6], in this study protocol, improvement sustainability is conceptualized as the continuation of change effectiveness after the active implementation phase.

Change implementation at the hospital level presents unique challenges due to the large scale and complexity of the organization. Most hospitals, as complex systems [7,8], comprise many layers of structures and processes, within which multiple departments are interdependent. Within this system, multiple groups of healthcare providers are interrelated across hierarchies [9,10], and they may have different ways to fulfill organizational and professional priorities.

Despite the recognized challenges, empirical evidence and theoretical underpinnings to support sustained improvement from hospital-based large-scale change are lacking. It is acknowledged in the literature that the longer-term success of large-scale improvement initiatives in hospital settings is seldom demonstrated [11,12,13]. Previous studies, which did document the outcome of large-scale change, reported that hospital-wide improvement initiatives infrequently sustained the improvements made [14,15]. An estimated success rate of change over the longer term has been reported as less than one in three [16,17].

From a theoretical perspective, there is a limitation in the existing change theories/models to sufficiently support improvement sustainability in hospital settings [9,18,19,20]. While organizational factors for achieving initial change success have been identified in healthcare settings [21,22,23,24,25], the sufficiency of those factors for sustaining improvement is not well understood. Studies have highlighted theoretical limitations in existing conceptual frameworks toward improvement that is comprehensively supported and sustained [26,27,28].

Lewin’s Unfreezing-Moving-Freezing model takes change from a development perspective, in which implemented change is consolidated and institutionalized through the Freezing phase [29,30]. As such, Lewin’s model demonstrates the potential to inform the sustainability of change effectiveness. However, the factors and processes of each phase, particularly Freezing, need to be more explicit to facilitate meaningful application of the model in complex settings [31,32,33], such as hospitals.

The dearth of empirical evidence and limited theoretical underpinnings to support sustained change effectiveness leads to a problem. Without sufficient evidence to support the longer-term success of larger-scale quality initiatives, initial implementation success may be short lived. In hospitals, such shortfalls could have a negative impact on the provision of safer and higher-quality care and may require repeated endeavors at additional cost when improvements are unsustainable.

Learning from previous change and translating the lessons into actionable strategies is central to achieving sustainable change outcomes and development [30,32,34]. Organizational learning involves a change in the organization’s knowledge that occurs through experience and is manifested in behavior embedded in individuals, routines, and memory systems [35]. From an organizational learning perspective, applications of the learning lead to changes in staff actions (i.e., error correction), processes (e.g., policy), and context (e.g., basic assumptions) within the organization [35,36]. The organizational learning approach provides a useful theoretical lens for sustaining change outcomes. For hospitals, this means that these learnings could help avoid similar mistakes and improve upon identified strengths in implementing planned change, such as organizational initiatives. However, insufficient opportunity to learn from experience is a major challenge to change success [32]. The low success rate of longer-term change effectiveness could be addressed by context-specific learning from change experience and using the lessons to inform current and future change strategies. Sustainable change in healthcare requires deliberate and proactive planning [37,38]. For hospitals, forming their own organizational knowledge that is grounded in learning and outcomes from large-scale change could contribute to the effective embedding of organizational initiatives with longer-term benefits.

A recent review [39] of empirical evidence has identified that sustained improvement from hospital-wide quality initiatives requires concerted efforts across hospital hierarchies (i.e., change actors) before, during, and after the active implementation phase. The review provides a comprehensive framework of interrelationships among the identified factors for improvement sustainability. The evidence-based framework includes the process aspects that resemble Lewin’s Unfreeze-Moving-Freezing model [29,30] and interplays between each aspect, which could be used to further expand the model for its use in hospital settings. However, the findings of this review indicate the need to examine improvement sustainment in real-world settings due to the context dependency of change implementation in the complex hospital environment [40,41]. Robust empirical research on large-scale change in various hospital settings will add to practice-based evidence [42]. Moreover, further research can bring a balanced focus on the end users, such as frontline staff, who determine the success of embedded change [43].

### Study Aims and Research Questions

We propose to examine improvement sustainability of two large-scale initiatives in an Australian tertiary hospital and translate the lessons into strategies for achieving sustained improvement from large-scale change in hospital settings. Specifically, Aim 1 is to examine retrospectively two quality improvement initiatives implemented across a hospital (i.e., hospital-wide), including the relevant organizational indicator outcome and the experience of the change actors involved in the initiatives. Aim 2 is to obtain lessons from the initiative implementations that can be translated prospectively into change strategies to design and support improvement sustainability.

We pose an overarching research question, “How is hospital-wide change embedded for sustained improvement in patient safety and quality?”, to address evidence gaps in the empirical and theoretical knowledge base. The following sub-questions will be addressed:(1)What are the enablers and barriers to sustaining the improvement?(2)How do relevant organizational records demonstrate the initiative outcome?(3)What factors in hospital-wide change implementation lead to sustained improvement?(4)What hospital-wide change processes enable sustained improvement?

## 2. Design and Methods

### 2.1. Study Design

We propose to conduct a single-site, multiple-case study with a mixed-methods approach to fulfill the study’s aims. The reporting elements of the mixed-method approach provided in this document have been informed by the reporting guideline for Mixed-Methods Research Manuscript Preparation and Review in health services management studies [44].

#### 2.1.1. Multiple-Case Study

Case studies are used to obtain richer and context-specific insights on a particular phenomenon [45,46], such as improvement sustainability. A single case may not reflect sufficiently the organizational variables within a complex hospital environment [8,9,10], by which to examine change implementation and improvement sustainability. Therefore, this study proposes to examine two real-world hospital-wide initiatives at the same hospital (Initiative #1, and Initiative #2). Each initiative will be treated as a unit of case and examined separately and sequentially (Initiative #1 then Initiative #2) to capture and fully immerse in the distinct experience and indicator outcomes of each case [46].

Two initiatives will be chosen based on the similarities in (1) change scale (i.e., hospital-wide), (2) change scope (i.e., multiple stakeholder groups), and (3) the aim of the initiative (i.e., to improve patient safety and quality). The study will be conducted in one hospital setting to ensure a consistent change context and eliminate variation introduced by different organizational settings that may influence the experience and indicator outcome of the initiatives. In addition, the quality-improvement oriented objective of the two initiatives will allow comparisons of the various change elements utilized [47]. This approach has been chosen because it will enhance rigor while ascertaining a variety of commonalities and differences among the cases in a single-site hospital context.

This study will select a completed initiative (Initiative #1) and another that is under implementation (Initiative #2) across the study site. This is to obtain actionable learnings from the first initiative and inform the strategies and actions of the second initiative’s implementation. Together, this approach will assist hospitals in a similar context to sustain any gains resulting from large-scale change initiatives.

#### 2.1.2. Mixed Methods

This study will adopt the one-phase (convergent) mixed-methods approach described by Creswell and Creswell [48], which involves the concurrent collection of separate sets of quantitative and qualitative data and integrating them for interpretation for each case. This approach will optimize the comprehensiveness and rigor of data collection and analysis to address the research questions. Integrated quantitative and qualitative data will provide stronger insights than a single type of data [48]. In a complex hospital environment [9], organizational indicators (i.e., quantitative data) alone may not fully reflect the outcome and process of the change implementation without considering the experience of the participants (i.e., qualitative data), and vice versa.

Together, this study will comprehensively capture the experience and relevant indicator outcomes from the two initiatives to inform change strategies for optimizing improvement sustainability. This approach, namely ‘Looking back, looking forward’ [49], is illustrated in Figure 1.

### 2.2. Study Setting

This study will be conducted at a public tertiary referral hospital in a regional area of Australia. The area is part of a state with a population of approximately 560,000 [50]. As a tertiary hospital, the study site is the referral center for patients who require intensive care and has extensive laboratory and clinical service facilities [51]. The hospital provides a range of general and specialty treatments ranging from acute, sub-acute, mental health, and inpatient aged care to approximately 250,000 people. It is a major clinical research and teaching facility in the area in affiliation with a university. The hospital is the largest referral center in the area, with approximately 63,000 admissions and 65,000 emergency department visits as of 2020. There are approximately 500 beds, employing around 3000 staff, and this facility is part of the broader public health service of the state.

### 2.3. Initiatives Selected for This Study

The two selected initiatives are Right Time Every Time (Initiative #1) and Speaking Up For Safety^TM^ (Initiative #2). These initiatives have been selected for (1) the change scale being hospital-wide, (2) the complexity involving multi-disciplinary stakeholder groups and various staff roles, and (3) the aims of both initiatives being improvement in patient safety and quality of care.

The scale of change and complexity of the selected initiatives are sufficiently representative of the phenomenon being examined (i.e., improvement sustainability of large-scale change), and this will allow rich data to address the research questions [46].

#### 2.3.1. Right Time Every Time (Initiative #1)

The aim of this initiative was to reduce incidences of medication omission, delay, and duplication across the study site. The initiative was implemented between June 2014 and June 2015. It included multiple interventions, such as developing and updating relevant protocols within the study site, a hospital-wide promotional campaign, and changes in the pharmacy systems for prescribing and dispensing prescribed medication. The compliance rate of omitted drugs within the National Inpatient Medication Chart (NIMC) Audit was used to measure the effectiveness of this initiative. The Audit is standardized and coordinated nationally by the Australian Commission on Safety and Quality in Health Care (ACSQHC), an Australian government agency that sets regulatory standards for safety, quality, and clinical care [52]. The NIMC Audit evaluated the safety and quality of paper-based medication charting in Australian hospitals and was replaced by the National Standard Medication Chart Audit in 2018 [52].

#### 2.3.2. Speaking Up for Safety^TM^ (Initiative #2)

The second initiative was launched in 2020 to implement the Speaking Up For Safety^TM^ program developed by the Cognitive Institute. This program aims to improve communication when escalating patient safety concerns to build a safety culture [53]. The program is being implemented across the broader region of the health service in the state. Currently, the study site is undergoing an ‘awareness phase’, during which all staff members are invited to attend a seminar presentation. The presentations are delivered by the service’s staff, who have been trained and licensed (called ‘Champions’) to provide education on this initiative. Presently, there is no agreed measurement by which to evaluate the effectiveness of the program. However, a routinely conducted organizational staff survey has been identified for utilization, as it includes questions relevant to the program’s intent and objective.

## 3. Study Procedures

The sequential approach to the multiple cases will involve examining Initiative #1, followed by Initiative #2. This approach will allow immersion in the unique experience and indicator outcomes of each initiative [45,46]. Each case will be examined through the data collected using the one-phase mixed-methods approach which will involve staff experience, routinely collected organizational indicator data and other organizational records, such as relevant organizational protocols [48]. A summary of the mixed-methods data, to be collected for this study is provided in Table 1.

### 3.1. Quantitative Data Collection—Routinely Collected Organizational Data

For Initiative #1, the annual medication omission rate within the NMIC Audit results for the period of 2012–2021 will be collected. The omission rate at a baseline (prior to the initiative implementation), during the initiative implementation, and in the post-implementation phase will be assessed. The audit tool was developed and managed by the ACSQHC for national use. This measure is consistent with that used during the initiative’s implementation in 2014–2015. The audit results are collected and published internally at the hospital level; therefore, no identifiable or personal information will be included in this dataset. The annual trend of the medication omission rate for the study site will be assessed to identify any rate changes during and after the implementation of the initiative. The audit results of the post-implementation years will inform the longer-term outcome of the initiative in relation to Aim 1.

For Initiative #2, the responses to selected questions within a routinely conducted Staff Engagement Survey will be used. Examples of the selected questions are “Employees in my work unit make every effort to deliver safe, error-free care” and “I can report patient safety mistakes without fear of punishment”. The selected questions indicate sufficient alignment with the aim of Initiative #2. Responders are asked to rate their agreement with these statements on a Likert scale of 1 (Strongly disagree) to 5 (Strongly agree). Survey participation is voluntary and conducted anonymously by a provider, Insync^®^, with an internally developed questionnaire. The survey is used among healthcare organizations in Australia and internationally, particularly for the external benchmarking feature provided by the survey company. At the study site, the survey commenced in 2018 for nursing staff and expanded in 2020 to include medical, allied health, and clinical support staff. The survey is conducted biennially. Changes in the ratings for selected questions with the introduction of Initiative #2 will be assessed as an interim outcome measure. Anonymized responses will be used, with no identifiable or personal information to be collected.

### 3.2. Qualitative Data Collection

We will conduct semi-structured, one-on-one staff interviews and collect organizational documents, such as relevant policies and internal reports, as the qualitative data collection method [48] to capture both individual experience and organizational evidence about the initiatives’ implementation.

#### 3.2.1. Semi-Structured Individual Staff Interviews

The first author will conduct semi-structured interviews with individual hospital staff members. Eligibility criteria include clinical or non-clinical staff from the study site who took part in Initiative #1 and those currently taking part in Initiative #2. Eligible staff will be invited to reflect on their experience with the relevant initiative. Semi-structured interviews can facilitate deep understanding by providing the opportunity to explore participants’ experiences and reflections on the initiatives [54]. Interviews will last approximately 45–60 min and be conducted in a quiet setting via video conferencing or in person at the study site or negotiated venue. Interviews will be recorded and transcribed verbatim with participant consent. A demographic questionnaire will be administered to capture the characteristics of the participants, including their professional background, years of employment, and previous education relevant to change implementation. Written informed consent will be obtained from all participants prior to the interview.

##### Participant Stratification

Implementing large-scale change in the complex hospital environment involves multiple groups of individuals professionally and hierarchically [4,9,10]. This complexity requires capturing diverse perspectives from different functions within the role hierarchy, such as senior managers, first-level managers, frontline staff, and the implementation team involved in the initiative’s implementation. Therefore, we will stratify participants into homogenous role groups to allow exploration of unique perspectives on initiative implementation and improvement sustainability across role variations (Table 2). Role stratification helps to capture the distinctive roles of the healthcare workforce at different levels of hierarchy to achieve and sustain improvement [4,39,55].

The total number of interview participants will be between 32 and 36 (Table 2). Data sufficiency can be reached using a relatively small number of participants who have higher “information power” [56], the degree to which participants can provide necessary insight to answer the research question based on their relevant experience [56]. This approximation considers the number of participants needed in each role/function for adequate capture of potentially distinctive experiences with the initiative’s implementation and to explore improvement sustainability.

##### Interview Questions

Interviews will explore staff experience of the implementation of the initiatives across three main domains—people, process, and organizational environment—and 11 subdomains, empirically evidenced as the factors to sustaining improvement [39]. Examples of subdomains that will inform the interview discussion are leadership, change management and perceived hospital culture supported for embedding the initiatives (Figure 2). Some interview questions will be customized for the stratified participant groups relevant to the role variation. For example, for frontline staff, there will be additional questions about perceived leadership at the unit and hospital levels. For the senior manager group, questions will explore more of their experience with the organizational embedding of the initiatives.

#### 3.2.2. Organizational Records

The electronic strategic document system and the Intranet within the study site will be utilized to search and retrieve relevant electronic organizational records relevant to the two initiatives. Strategic documents, such as policies and protocols, will be searched to identify any outputs or evidence of the organizational embedding of the initiatives. Operational documents such as reports produced from the initiative and staff communications such as newsletters and memos will be obtained to capture the activities and endorsed messages from the hospital management regarding the initiatives.

### 3.3. Data Analysis

#### 3.3.1. Multiple-Case Analysis

Each initiative, as a case, will be analyzed separately (i.e., within a case) and comparatively (i.e., across cases) to identify similarities and differences [45,47,57]. The observed themes and outcomes evidenced by the merged and interpreted data through mixed methods [48] will address the study aims and provide a fuller picture of change implementation and improvement sustainability within the study site (Figure 1).

#### 3.3.2. Quantitative Data Analysis

Descriptive statistical analysis will be conducted on the numerical data collected: (1) the omitted medication audit results (Initiative #1), (2) the selected survey responses (Initiative #2), and (3) the demographic questionnaire responses. The statistical analysis will address Aim 1 by quantitatively summarizing measurable hospital indicator outcomes relevant to the two initiatives and the characteristics of interview participants. The analysis for the first two data items will be a time series (trend) analysis of the indicator outcome before, during (Initiative #1 and #2) and after (Initiative #1) the initiative’s implementation. Microsoft Excel (version 2302) or IBM SPSS Statistics V.29.12 will be used for the time series analysis. The time series analysis will assess whether the implementation affected the indicator outcome and whether the impact of Initiative #1 was sustained. For Initiative #2, staff survey responses prior to the initiative introduction will be compared with those after the introduction. Demographic data from the interview participants will be analyzed through Research Electronic Data Capture (REDCap), a secure web-based research data capture tool [58].

Although the statistical analyses will examine potential long-term (Initiative #1) and interim (Initiative #2) impacts of the intervention, caution will be taken that the initiatives will not be the sole factor in any numerical changes in the indicator outcome and that there may be a variety of confounding factors. Accordingly, the statistical analysis is not expected to identify the causal factors for the changes. However, inferential statistical analysis may be conducted to identify correlations within the staff survey responses, such as a relationship between other questionnaire items and the selected items relevant to patient safety.

#### 3.3.3. Qualitative Data Analysis

The qualitative data from interviews and organizational documents will be analyzed to address Aim 1 and Aim 2 by exploring the participant experience and relevant organizational documents from the perspective of improvement sustainability. Data collected through interviews and free-text responses from the previously conducted surveys will be analyzed using the thematic analysis method elaborated by Nowell et al. [59]. This method involves: (1) familiarizing oneself with the collected data; (2) generating initial codes; (3) searching for themes; (4) reviewing themes; (5) defining and naming themes; and (6) producing the report. Such a structured approach will allow examining different perspectives through similarities and differences captured in the data [59]. A coding framework will be developed and aligned with the interview question themes [39]. The stratified participants will be assigned a unique code to be indirectly identified in separate groups (Table 3). For example, a doctor who participated in Initiative #1 without managerial responsibilities will be coded as 1G4DR, and a Nurse Unit Manager who is participating in Initiative #2 will be coded as 2G3NR.

The collected organizational documents, such as protocols and staff communication, will be examined through document analysis, which involves (1) reading the material, (2) extracting data, (3) analyzing data and (4) distilling the findings [60]. This approach supports rigor through systematic and practical guidance for evaluating and analyzing organizational documents.

These qualitative analyses will occur concurrently with the data collection. This approach will allow the interview questions to be refined based on findings that emerge from the interviews [61]. NVivo software (Release 1.7, version 1533) [62] will be used to manage qualitative data.

#### 3.3.4. Integrating Mixed-Methods Data

The mixed-methods datasets for each case will be integrated by merging [63] within the one-phase (convergent) approach [48] (Figure 1). The integrated data will provide a triangulated and synergetic overview of the improvement sustainability of Initiatives #1 and #2 within the study site [48,64]. The findings of the integrated data for each initiative will be compared, and divergence and inconsistencies between the datasets will be identified [44]. The described analysis approach has been adopted to allow sufficient rigor through rich mixed-methods data from two different initiatives [65]. The findings obtained will address Aim 1.

Further, lessons emerging from the qualitative datasets relevant to improvement sustainability will be synthesized. These will be translated into actionable strategies to guide large-scale change with sustained improvement within the study site and potentially for broader application in similar settings. The synthesized actions will be informed by the triangulated approach consistent with the above. This approach has been chosen for the comprehensive, context-driven evidence necessary to sufficiently address Aim 2.

Together, the integration of the data is designed to draw a fuller picture of the improvement sustainability of large-scale change within the study site as well as utilize lessons to inform improvement sustainability. This approach is chosen as a recognized strategy to embed large-scale change through learning from experience in context [32,35].

## 4. Discussion

By addressing the research questions, this study aims to comprehensively examine the past and current large-scale quality initiatives in a tertiary hospital and obtain actionable knowledge for attaining sustained improvement from large-scale change in comparable settings. This study seeks to contribute relevant knowledge for practice and theoretical underpinnings to support organizational development through sustained improvement from change efforts.

For the study site, information obtained from the research questions will provide evidence to develop site-specific organizational knowledge and learning in improvement sustainability of large-scale change [30,32,34,35]. The advanced knowledge, based on evidence from the local context, has the potential to assist hospitals in optimizing time and resource investment for large-scale change and sustaining these outcomes. Previous studies have shown that the organizational context determines the implementation outcome [23,66]. The information gained through the sub-questions will provide context-based evidence that may demonstrate actionable strategies to consistently deliver safer and higher quality care. Learning from change experience is a recognized opportunity toward successful and meaningful organizational change [7,32] and an essential step to transforming organizational experience into knowledge [35]. The ability to learn in such a way is a vital element in adapting to the dynamic healthcare landscape [67].

The study findings will add to the evidence base to support longer-term improvement from large-scale change in hospital settings. The multiple-case study design allows comprehensive reflections of organizational variables within the complex environment of hospitals [8,9,10] that may be unique to other settings such as care homes [68]. Particularly, the change outcomes of and the participant experience with Initiative #1 will address calls for longer-term change evaluation [11,12,13]. The organizational learning approach of prospectively translating the lesson from previous change to current and future change will contribute to the evidence base to inform change strategies for longer-term effectiveness [32]. Additionally, improvement sustainability from the perspectives of the participants could add to the change management literature that is traditionally driven by managerial viewpoints [32]. The value of understanding change participants’ views and their interpretation of change as a recognized opportunity for successful and meaningful change [32,69]. The participant interviews in this study will provide a lens to evaluate organizational change through the examined initiatives and the ability to sustain the benefits of the change.

From a theoretical perspective, the study findings will present the potential to advance the theoretical underpinnings to support and design organizational improvement sustainability in hospital settings. The knowledge obtained of the factors that facilitate or constrain improvement sustainability will provide organizational variables and/or determinants in large-scale change. These factors could be as considered as potential organizational determinants in conjunction with those included in frameworks that are widely used in healthcare [70,71] to enhance improvement sustainability in change implementation. The identified elements of change processes that lead to sustaining improvement will offer evidence to guide organizational change in hospital settings or be incorporated within current change process models to elaborate on improvement sustainability. Lewin’s Unfreezing-Moving-Freezing model [29,30] provides a flexible platform that is well established in healthcare [33]. Available evidence has identified a high affinity between Lewin’s model and the review findings of the change process that enabled sustained improvement [39].

The proposed study has strengths and limitations. The mixed methods and multiple-case study design of this study cater for the complexity that is inherent to the large hospital environment [7,8,9,10]. Those approaches enable capturing a fuller picture of change implementation and improvement sustainability in the study organization through triangulating organizational records and participant experience. Mixed methods will allow unique and balanced perspectives on interim and longer-term outcomes from the initiatives and their improvement sustainability. The stratified cohorts for interview participants are designed to ascertain the holistic perspectives of the participants involved in the examined initiatives across organizational hierarchies and various disciplines. This structure is reflective of the intricate layers of contemporary healthcare organizations [4,55]. Improvement sustainability seen through the lens of participant experience across different role positions will offer knowledge that may be of actionable value for hospital managers and health service researchers. Therefore, this proposed study could be appropriate for audiences across different healthcare disciplines in complex settings where change is introduced. Limitations of this study include reduced generalizability due to the single-site study design, although the findings will provide a rich organizational knowledge base for the study site to develop strategies for sustaining improvement from change. A relatively small sample size for interview participants is another limitation of this study, which aims to capture an organizational view. Stratifying participant groups to capture various viewpoints across organizational hierarchy and disciplines is a mitigation strategy to prevent misrepresenting an organizational view from a single group.

## 5. Conclusions

This study aims to obtain a comprehensive organizational view of improvement sustainability through large-scale initiatives and translate the lessons into actionable strategies for achieving sustained improvement in hospital settings. The multiple-case study design with mixed methods will allow rich data to comprehensively address evidence gaps and produce knowledge that can be applied in practice and research settings. Examining past and current initiatives, this study will generate lessons for guiding organizational change in which the gained improvement can be maintained in longer-term. We seek to optimize the outcome of the efforts and investment for organizational change toward improved patient care outcome in complex hospital settings.

## 6. Dissemination

Anonymized results of this study will be disseminated locally and internationally through peer-reviewed journals, conference presentations, and a doctoral thesis (SEJM). The stakeholders of the study site will be shared with the study results in a variety of forms, such as reports and presentations.

## Figures and Tables

**Figure 1 healthcare-11-02175-f001:**
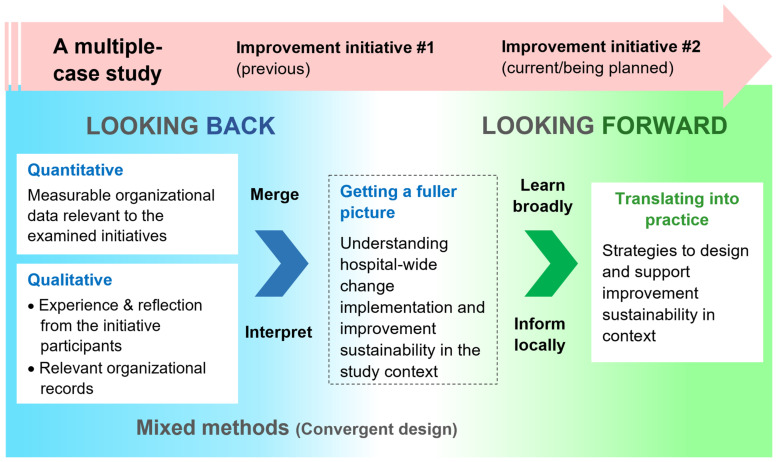
Illustration of the research design—looking back and looking forward—a single-site, multiple-case study [45] involving a convergent mixed-methods approach [48].

**Figure 2 healthcare-11-02175-f002:**
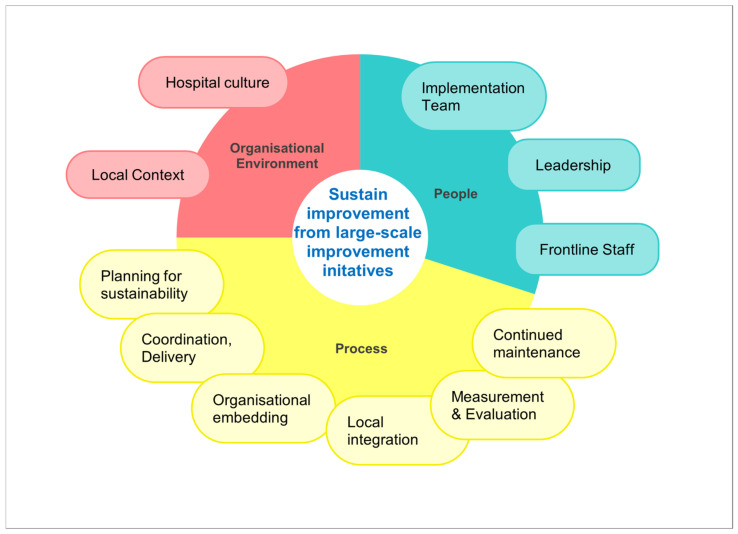
Domains of factors for sustained improvement from hospital-wide initiatives. Adapted from Moon et al. [39].

**Table 1 healthcare-11-02175-t001:** Proposed mixed-methods data collection.

Mixed-Methods Data	Initiative #1 (2014–2015)	Initiative #2 (2020–Present)
**Quantitative data**: Routinely collected organizational data	National Inpatient Medication Chart Audit results on omitted medication (2012–2019)	Selected questionnaire items relating to patient safety from Staff Engagement Surveys (2018–latest)
	A demographic questionnaire accompanying the interview	A demographic questionnaire accompanying the interview
**Qualitative data**: Interviews and organizational records	Semi-structured interview with involved staff	Semi-structured interview with involved staff
	Organizational records relevant with the initiative (e.g., Report, policy)	Organizational records relevant with the initiative (e.g., Report, policy)

**Table 2 healthcare-11-02175-t002:** Target number of interview participants by stratified group and initiative.

Participant Group	Initiative #1	Initiative #2	Total
Group 1: Senior managers	3–4	5	8–9
Group 2: Implementation team members	3–4	5	8–9
Group 3: Frontline managers	3–4	5	8–9
Group 4: Frontline staff	3–4	5	8–9
Total Interview participant no.	12–16	20	32–36

**Table 3 healthcare-11-02175-t003:** Coding scheme for the interview participants.

Initiative Code	Participant Group Code	Profession Code
Initiative #1: 1	Group 1: G1	Nurse: NR
Initiative #2: 2	Group 2: G2	Doctor: DR
	Group 3: G3	Allied Health: AH
	Group 4: G4	Non-clinical professionals: NC

## Data Availability

Not applicable.

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
