# Peer review of "Looking Back, Looking Forward: A Study Protocol for a Mixed-Methods Multiple-Case Study to Examine Improvement Sustainability of Large-Scale Initiatives in Tertiary Hospitals"

_healthcare, 2023, doi:10.3390/healthcare11152175_

Round 1
Reviewer 1 Report
The research has an exciting topic and promising significance. Putting aside the background and research design, my biggest concern is that the manuscript is not a paper. It is a proposal. Without actual findings, it is unlikely that this manuscript will bring actual contribution to our knowledge. I believe that once finished, this manuscript should be of the interest of many authors. I would like to see the authors' thoughts/responses to that and their future plans regarding completing thsi research.
Author Response
Response to Reviewer 1 Comments
Point 1: The research has an exciting topic and promising significance. Putting aside the background and research design, my biggest concern is that the manuscript is not a paper. It is a proposal. Without actual findings, it is unlikely that this manuscript will bring actual contribution to our knowledge. I believe that once finished, this manuscript should be of the interest of many authors. I would like to see the authors' thoughts/responses to that and their future plans regarding completing this research.
Response 1:
We thank you for your positive feedback on the topic and potential significance of our research.
Protocol is one of the article types accepted by the Journal (MDPI Healthcare) for publication which is introduced in the Journal’s website.
Study protocols are actively published in the journal. Examples of recent publications are protocols by Karkauskiene et al (2023), Garay-Sanchez (2023) and Cabo et al (2022).
In line with the Journal’s instruction for protocols and the previously published protocols such as above, we have provided a detailed description of the research design and methods step-by-step within the instructed manuscript structure, upholding academic rigor and based on evidence as well as considering study feasibility in the study context.
We believe that this manuscript meets the Journal’s requirements.
Once analysis has been completed, we will submit manuscripts to report the findings from this study. We hope to introduce you and the research/practitioner communities to the study findings through research articles soon.
Reviewer 2 Report
This article raises a very interesting and important topic assessing the impact of patient safety and quality improvement initiatives at the organizational level on long-term performance improvement and sustainable development.
The article is in line with the profile of the journal, relevant to the field and presented in a structured way, but improvements should be made in the text of the article:
1) The authors have not clearly formulated the research problem. Please emphasise in the introduction why your research is important from a theoretical and practical point of view. What research gap are you trying to fill and what theory do you intend to develop?
2) The authors formulate the objectives (Lines 101-105):
One – “to retrospectively examine two quality improvement initiatives implemented across a hospital (i.e., hospital-wide), including the relevant organizational indicator outcome and the experience of the change actors involved in the initiatives.
Aim 2 is to obtain lessons from the initiative implementations that can be prospectively translated into change strategies to design and support improvement sustainability”
Comments: The objectives of the study should arise from the research gaps and problems indicated in the introduction. So please elaborate on the theoretical part in the introduction.
The authors formulated the following research question: “How is hospital-wide change embedded for sustained improvement in patient safety and quality?”
Comments: Isn't this question too general? Maybe you could try to formulate 2-3 questions more precisely?
3) Please add to the discussion more results of other studies and refer to the research questions. Discussion is too short.
4) Conclusion is missing. In this section should be formulated general conclusions of the study, limitations, implication and directions for future research.
5) The figures and tables are appropriate.
Author Response
Response to Reviewer 2 Comments
Point 1: This article raises a very interesting and important topic assessing the impact of patient safety and quality improvement initiatives at the organizational level on long-term performance improvement and sustainable development.
The article is in line with the profile of the journal, relevant to the field and presented in a structured way, but improvements should be made in the text of the article:
1) The authors have not clearly formulated the research problem. Please emphasise in the introduction why your research is important from a theoretical and practical point of view. What research gap are you trying to fill and what theory do you intend to develop?
Response 1: We thank you for your positive feedback on the topic and relevance of our research.
We agree that there can be more clarity. In response to your comments, we have made substantial revisions in the Introduction. The revisions have been marked in yellow highlight as per the Journal’s instruction.
Please, refer to Lines 56-77 and 103-106 of the manuscript to view these amendments.
Point 2: 2) The authors formulate the objectives (Lines 101-105):
One – “to retrospectively examine two quality improvement initiatives implemented across a hospital (i.e., hospital-wide), including the relevant organizational indicator outcome and the experience of the change actors involved in the initiatives.
Aim 2 is to obtain lessons from the initiative implementations that can be prospectively translated into change strategies to design and support improvement sustainability”
Comments: The objectives of the study should arise from the research gaps and problems indicated in the introduction. So please elaborate on the theoretical part in the introduction.
Response 2: The above revisions include elaborations of the theory underpinning this study. Please, refer to the highlighted lines in the Introduction section in Lines 56-77 and 103-106.
Point 3: The authors formulated the following research question: “How is hospital-wide change embedded for sustained improvement in patient safety and quality?”
Comments: Isn't this question too general? Maybe you could try to formulate 2-3 questions more precisely?
Response 3: We agree that the overarching question alone may not be sufficiently specific. Therefore, we have included 4 sub-questions to address the overarching question. We also have separated the study aim and research questions in a new sub-section to make these clear. Please refer to Lines 112-129.
Point 4: 3) Please add to the discussion more results of other studies and refer to the research questions. Discussion is too short.
Response 4: We agree that the discussion could be further elaborated. In response, we have made an extensive revision and addition in the section including anticipated contributions to practice and theory as well as the strengths and limitation of this study. Please refer to Lines 393-459.
Point 5: 4) Conclusion is missing. In this section should be formulated general conclusions of the study, limitations, implication and directions for future research.
Response 5: We have added a new section for the conclusion as per your feedback, containing a short summary of the study, anticipated implications of the findings appropriate to a study protocol. Please refer to Lines 460-469.
We have included strengths and limitations in the discussion section for better flow and balance for the readers following the anticipated implications of the study findings.
We will address directions for future research based on the evidence obtained through the study findings. We think that this approach will allow us to provide more informed and impactful directions.
Point 6: 5) The figures and tables are appropriate.
Response 6: Thank you for this endorsement.
Reviewer 3 Report
Dear Authors
I have a few remarks on your text.
Minor
1. Please change the future tense in Abstract to the present. Composing an abstract means you convey what you have done and not what you will do, in spite of you may describe a new protocol.
2. Your current abstract has too many common statements. I recommend adhering to Introduction–Methods–Results–Conclusions scheme, though without mentioning corresponding words. Please be more specific in your Abstract about your findings.
3. Lines 36, 101, etc. Please eschew using split infinitives. Though admissible in spoken language, it is a grammar violation in classic modern English.
4. Lines 106–107: please make your major research question more specific and understandable for a common reader untrained in medical organisation management.
5. Line 109: “We propose to conduct…”? Do not be estranged from yourself in every phrase.
6. Fig. 1 seems to have insufficient resolution, at least in the PDF version of your text I am reading. Please be sure you have supplied both figures with at least 600dpi resolution.
7. Line 159. Do not be enigmatic. Place yourself into a reader’s situation. He or she must not solve riddles, to be sure. Which state do you mean?
8. Line 158. Not all countries use primary–secondary–tertiary healthcare classification. Please explain the term “tertiary referral hospital.”
9. Please move “Ethics” section to the end of your text.
Major
1. As I understood, your major goal is to compose a research protocol for a future study in a chosen hospital. If so, it would be advisable to disclose which Australian hospital you will work in and to expound why the hospital’s structure and operations potentially allow implementing your new research protocol. This has to be written overtly in Introduction and discussed afterwards. As for now, your true purpose is obscure.
After the improvement your text may be published.
Thank you.
Best regards,
the Reviewer
Author Response
Response to Reviewer 3 Comments
Point 1: Please change the future tense in Abstract to the present. Composing an abstract means you convey what you have done and not what you will do, in spite of you may describe a new protocol.
Response 1: We have followed the examples of the study protocols recently published in the Journal, such as those by Karkauskiene et al (2023), Garay-Sanchez (2023) and Cabo (2022).
In these protocols future tense is used to describe what will be done as per the methods described. We have reviewed our abstract and used present tense for the elements of background, rationale and aim. We then used future tense for what we will be doing in the senses to describe the research methods, analysis and dissemination. We have yet to commence on data collection.
Point 2: Your current abstract has too many common statements. I recommend adhering to Introduction–Methods–Results–Conclusions scheme, though without mentioning corresponding words. Please be more specific in your Abstract about your findings.
Response 2: In response to your comment, we have revisions in our abstract to include more specific aspects of this study.
As the Journal’s instruction for abstract is not restrictive to a particular scheme, we took the liberty of modifying your suggested structure: Background, Aim, Design & Methods and Implications.
Point 3: Lines 36, 101, etc. Please eschew using split infinitives. Though admissible in spoken language, it is a grammar violation in classic modern English.
Response 3: In response to your feedback, phrasing and grammar have been amended throughout the manuscript. Focusing on split infinitives, we have attempted simplifying language.
Point 4: Lines 106–107: please make your major research question more specific and understandable for a common reader untrained in medical organisation management.
Response 4: We agree that the overarching question alone may not be sufficiently specific. Therefore, we have included 4 sub-questions to address the overarching question. We also have separated the study aim and research questions in a new sub-section to make these clear. Please refer to Lines 112-129.
Point 5: Line 109: “We propose to conduct…”? Do not be estranged from yourself in every phrase.
Response 5: We agree that the language can be refined to be less passive. We have changed the line as your suggestion: “We propose to conduct a single-site multiple-case study with a mixed-methods approach to fulfil the study aims.” (Line 132).
Additionally, passive phrasing has been amended throughout the manuscript where appropriate.
Point 6: Fig. 1 seems to have insufficient resolution, at least in the PDF version of your text I am reading. Please be sure you have supplied both figures with at least 600dpi resolution.
Response 6: The two figures have been replaced with images in Tag Image File Format and meet the resolution requirement of 300dpi or higher by the Journal.
Point 7: Line 159. Do not be enigmatic. Place yourself into a reader’s situation. He or she must not solve riddles, to be sure. Which state do you mean?
Response 7: The authors are concerned that there is a risk to the study site of being easily identifiable because there are few acute hospitals in the area (State). We also anticipate that further consequence is the high possibility of jeopardising the confidentiality of the study participants.
As we approach these risks cautiously and respect the privacy of the study site and the participants, we therefore have agreed that we will not disclose the name of the study site.
However, we will seek an opportunity to discuss with relevant stakeholders of the study site when the research findings are distributed to them for their consideration for disclosing the site name.
We have addressed the above risks in the manuscript (Lines 477-479) to inform readers.
Point 8: Line 158. Not all countries use primary–secondary–tertiary healthcare classification. Please explain the term “tertiary referral hospital.”
Response 8: We agree that providing a clear scope by what tertiary care means within the context of the study would help those in different healthcare systems interpret the findings. We have clarified the term tertiary hospital in th section of Study Setting (Lines 179-181) as follows and used a reference to support:
A tertiary hospital is the referral center for patients who require intensive care, such as life support, with extensive laboratory and clinical service facilities.
Point 9: Please move “Ethics” section to the end of your text.
Response 9: We thank the reviewer for picking up this oversight. We have relocated the ethics section to the end of the manuscript body (Lines 475-484) in response to your feedback.
Point 10: As I understood, your major goal is to compose a research protocol for a future study in a chosen hospital. If so, it would be advisable to disclose which Australian hospital you will work in and to expound why the hospital’s structure and operations potentially allow implementing your new research protocol. This has to be written overtly in Introduction and discussed afterwards. As for now, your true purpose is obscure.
Response 10: Please see our response above for Point 7 in which we have detailed the risks to the study site and the participants that could be incurred following the suggested disclosure.
We thank you for your constructive feedback.
Round 2
Reviewer 1 Report
The authors have addressed the issues sufficiently.
Author Response
Thank you very much for your positive feedback.
Reviewer 2 Report
Dear authors,
thank you for the corrections made to the article.
The article contains the necessary elements, is interesting for the reader, and helpful for making changes resulting in sustainable development.
All my comments from the review have been implemented, the article should be published.
Author Response
Thank you very much for your positive, constructive feedback.